# Young at risk-people in Maputo City, Mozambique, present a high willingness to participate in HIV trials: Results from an HIV vaccine preparedness cohort study

**Igor P. U. Capitine**[1,2]*, **Ivalda B. Macicame**[1], **Artur M. Uanela**[1], **Nilesh B. Bhatt**[1], **Adam Yates**[3,4], **Mark Milazzo**[3,4], **Chiaka Nwoga**[3,4], **Trevor A. Crowell**[3,4], **Nelson L. Michael**[3], **Merlin L. Robb**[3,4], **Ilesh V. Jani**[1], **Arne Kroidl**[2,5,6], **Christina S. Polyak**[3,4], **Caroline De Schacht**[7], **for the RV363 Study Group**[¶]

1 Instituto Nacional de Saúde (INS), Maputo Province, Mozambique, 2 Centre for International Health (CIH), University of Munich (LMU), Munich, Germany, 3 U.S. Military HIV Research Program, Walter Reed Army Institute of Research, Silver Spring, Maryland, United States of America, 4 Henry M. Jackson Foundation for the Advancement of Military Medicine, Bethesda, Maryland, United States of America, 5 Division of Infectious Diseases and Tropical Medicine, Medical Center of the University of Munich (LMU), Munich, Germany, 6 German Center for Infection Research (DZIF), Munich, Germany, 7 Friends in Global Health, Maputo, Mozambique

¶ Membership of the RV363 Study Group is provided in the acknowledgments
* lcapitine@gmail.com

**Data Availability Statement:** The dataset supporting the conclusions of this article is

## Abstract

### Introduction

Vaccine efficacy testing requires engagement of willing volunteers with high disease incidence. We evaluated factors associated with willingness to participate in potential future HIV vaccine trials in Maputo, Mozambique.

### Methods

Adults aged 18–35 years without HIV and who reported at least two sexual partners in the 3 months prior to screening were enrolled into a 24-month observational study. They were asked at screening and exit if they would be willing to participate in a theoretical HIV vaccine study. Bivariate and multivariate logistic regression analyses were done between willingness to participate, demographic, sexual behavior, and motivational factors for screening visit data. Logistic regression with generalized estimating equations (GEE) was used to estimate odds ratios (ORs) and 95% confidence intervals (CIs) for factors potentially associated with willingness to participate for data from both visits.

### Results

A total of 577 participants without HIV were eligible, including 275 (48%) women. The mean age was 22.2 (SD ± 3.9) years. At screening 529 (92%) expressed willingness to participate and the proportion remained stable at 378 (88%) of the 430 participants retained through the exit visit (p = 0.209). Helping the country (n = 556) and fear of needles (n = 26) were the

available in the Harvard Dataverse repository: https://doi.org/10.7910/DVN/RBTW80.

**Funding:** This work was supported by a cooperative agreement (W81XWH-07-2-0067) between the Henry M. Jackson Foundation for the Advancement of Military Medicine, Inc., and the U. S. Department of Defense (DOD). This research was funded, in part, by the U.S. National Institute of Allergy and Infectious Diseases. Material has been reviewed by the Walter Reed Army Institute of Research. There is no objection to its presentation and/or publication. The opinions or assertions contained herein are the private views of the author, and are not to be construed as official, or as reflecting true views of the Department of the Army or the Department of Defense. The investigators have adhered to the policies for protection of human subjects as prescribed in AR 70–25.

**Competing interests:** The authors declare that they have no known competing financial interests or personal relationships that could have appeared to influence the work reported in this paper.

top motive and barrier for willingness to participate, respectively. Results from the GEE binary logistic regression (screening visit and exit visit) showed that wanting to learn how to avoid risk behaviors (aOR 3.33, 95% CI: 1.61–6.86) and feeling protected against HIV infection (aOR 2.24, 95% CI: 1.07–4.7) were associated with willingness to participate in HIV vaccine studies.

## Conclusion

The majority of our study population in Mozambique expressed willingness to participate in a theoretical HIV vaccine trial. Participation in a HIV vaccine trial was seen as a way to contribute to the fight against HIV but was associated with some unrealistic expectations such as protection against HIV. This reinforces the need for continuous mobilization and awareness of potential participants to HIV vaccine trial.

## Introduction

The burden of the HIV epidemic in Mozambique, a country with almost 28 million inhabitants, is high [1]. HIV prevalence in Mozambique's capital, Maputo City is 21.7% for women and 11.0% for men, aged 15–49 years [2].

Vaccines are considered one of the primary strategies for preventing infectious diseases. An effective HIV vaccine would prove a turning point on the path to achieving zero new HIV transmissions in Mozambique [3–5]. Mozambique has successfully conducted two HIV vaccine clinical trials, a phase I trial and a phase II trial, using HIV-DNA as a prime and modified vaccinia virus Ankara (HIV-MVA) as a boost [6,7], and is currently conducting additional vaccine trials [8] in a global effort to develop an HIV vaccine that is safe, affordable and efficacious for different populations [9].

A wide range of social and behavioral frameworks have been designed to improve the implementation of HIV preventive studies, resulting in a consensus that emphasis should be placed on socio-behavioral components, including the assessment of willingness to participate (WTP) in HIV vaccine studies [10–14].

Literature reviews have reported different hypothetical willingness to participate results from studies conducted in the same and/or different places, with the same and/or different populations. A myriad of factors such as safety of experimental HIV vaccines, distrust of those involved with clinical research, and risk of discrimination because of participation in HIV vaccine studies, have been identified as key factors influencing hypothetical willingness to participate. However, the importance of these factors also varies over time, populations, and contexts. Also important is the fact that, hypothetical willingness to participate, does not guarantee actual participation in a vaccine trial [15,16].

Identifying new factors and exploring underlying factors that can influence willingness to participate can help to explain the differences observed in willingness to participate in different communities [10,13]. This prompts for the need to continually evaluate whether individuals at risk for HIV infection would be willing to participate in HIV vaccine trials and to identify factors that could serve as barriers or facilitators for trial participation in a detailed and contextualized manner [17,18].

Relatively few willingness to participate studies have been conducted in low-income countries in Africa, including Mozambique. To our knowledge, there are no reports of willingness

to participate studies conducted in Maputo city, which have assessed willingness to participate over a period of time with the same participants [15,16,19]. Therefore, we evaluated factors associated with willingness to participate in future HIV vaccine trials during a two-year HIV incidence cohort study in Maputo City, Mozambique.

## Methods

### Study design and population

The RV363 study was a prospective observational HIV-incidence cohort study. It was conducted between November 2013 and October 2017 at the *Centro de Investigação e Treino em Saúde da Polana Caniço* (CISPOC), a health research and training center of Instituto Nacional de Saúde (INS). CISPOC has experience in conducting HIV vaccine clinical trials and is located in a populated low-income suburban area of Maputo city, Mozambique [8]. Inclusion criteria for the cohort study were age 18–35 years and self-report of at least two sexual partners in the 3 months prior to screening. HIV status was assessed using the Alere Determine™ HIV-1/2 rapid test (Alere Inc, Waltham, USA) with reactive tests confirmed using the Uni-Gold™ Recombigen® HIV-1/2—for HIV 1 and 2 (Trinity Biotech, Bray, Ireland). Women of childbearing potential were required to have a negative urine pregnancy test. Co-enrollment in any other HIV prevention study was not allowed. Participants who were eligible for the cohort study, were without HIV, and who answered the survey questions about willingness to participate in future HIV vaccine trials were included in these analyses.

### Sample size

The sample size was calculated considering a hypothetical incidence of at least 3% and prevalence of 16.1%, and a response rate of 50%. The expected retention rate was 85% over a 2-year follow-up period.

### Study procedures

Participants were evaluated every 3 months for up to 24 months, with HIV testing and extensive questionnaires as previously described [20]. Participants were required to pass a test of comprehension to confirm their understanding of the study procedures before signing the informed consent form. Paper-based questionnaires were administered to each volunteer by a trained staff member (face-to face interviews) at screening and at the exit visit 24 months later. Questionnaires collected data on participant demographics, sexual behaviors, and willingness to participate in future HIV vaccine trials.

### Questionnaires

**HIV vaccine knowledge.** Participants were asked "*if a vaccine is meant to prevent illness*" (true/false) and if they had received information on HIV research (yes/no) prior to their participation in the study, including the source of information.

**Willingness to participate questionnaire.** First, the staff member would give a brief explanation of the purpose of HIV vaccine clinical trials to the participant before proceeding to the main question, "*would you be willing to participate in such a study to test an experimental HIV vaccine*?" (yes/no/someone else should decide/refused to answer/don´t know). Next, participants had to indicate factors that could influence participation in a HIV vaccine study. Participants were asked to select barriers for not wanting to participate in a HIV vaccine trial, and to select personal benefits of participating in a study of an HIV vaccine. Facilitators for participation in a HIV vaccine study were assessed with the question "*how likely or unlikely would*

*you be to enroll in a research study of a new experimental HIV vaccine*", and responses were collected on a 4-point Likert scale: very unlikely, somewhat unlikely, somewhat likely, and very likely. Motives for wanting to participate in a HIV vaccine study were assessed with the question "*how important are the following factors in making a decision about participating in a research study of an experimental HIV vaccine*", with responses also collected on a 4-point Likert scale: not important at all, somewhat important, important, and very important. The last question was if the participant thought that an HIV vaccine would be useful in controlling HIV infection (yes/no). Participants could select all applicable answers for questions with more than one option. The questionnaire used can be found as S1 Table.

## Statistical analyses

Demographic variables (age, gender, level of education, marital status, income, main occupation, history of blood transfusion, desire to get pregnant) [16,21] and sexual behavior variables (age of first sexual intercourse and perceived risk for HIV infection) [22,23] included in the analyses were selected based on literature review. Expressed Willingness to Participate was the dependent variable and was determined by a "yes or no" question, with "do not know" and "refused to answer" as the other options. Only 17 participants chose "do not know" as an option and 0 participants chose "someone else should decide" or "refused to answer", but these were removed for bivariate and multivariate analysis. Likert scale questions were dichotomized from a set of 4 categories: "unlikely" (questionnaire responses "very unlikely" and "somewhat unlikely") and "likely" (questionnaire responses "somewhat likely" and "very likely"); "not important" (questionnaire responses "not important at all" and "somewhat important") and "important" (questionnaire responses "important" and "very important"). Frequencies and proportions for all variables were calculated. Two analytical approaches were employed, based on the status of the participants' visit assessments. For the screening assessment, bivariate analysis was done between willingness to participate (yes/no), demographic, sexual behavior, and willingness to participate factors. Differences were assessed with Chi-square and Fisher's exact test when appropriate. Willingness to participate factors that were statistically significant ($p<0.05$) in unadjusted logistic regression models were included in the adjusted logistic regression model at the baseline assessment. Analyses were performed among participants who stayed throughout the study, to identify factors associated with willingness to participate. McNemar's exact test was used to assess the differences in responses within the participants, at screening and at the exit visit, and variables with $p<0.05$ were included in a generalized estimating equation (GEE) logistic regression for binary outcomes model. Analyses were performed using STATA 15.1 (StataCorp, College Station, TX).

## Ethical approval

This study was approved by the National Health Bioethics Committee of Mozambique (IRB00002657), and by the Institutional Review Board Ethics Committee of the Walter Reed Army Institute of Research (IRB00000794). Administrative approval was granted by the Ministry of Health of Mozambique.

## Results

### Demographic and sexual behavioral characteristics

From 1150 screened volunteers, 577 were eligible for inclusion in this analysis, all of whom responded to the question about willingness to participate in a future HIV vaccine trial. Their characteristics are shown in Table 1. Briefly, 275 (48%) were female, the mean age was 22.2

**Table 1. Selected demographic and sexual behavior characteristics of 577 eligible HIV negative participants by willingness to participate at screening in a HIV-incidence cohort study (RV363 study).**

| | Willingness to Participate (N = 577) | | |
|---|---|---|---|
| | **No** | **Yes** | **Do not Know** |
| Age (categorized), n (%) | | | |
| *18–20* | 14 (2) | 225 (39) | 2 (0) |
| *21–24* | 14 (2) | 191 (33) | 5 (1) |
| *25–35* | 10 (2) | 113 (20) | 3 (1) |
| Gender | | | |
| *Male* | 23 (4) | 272 (47) | 7 (1) |
| *Female* | 15 (3) | 257 (45) | 3 (1) |
| Highest level of Education attained, n (%) | | | |
| *≤ Primary school completed* | 3 (1) | 54 (9) | 0 (0) |
| *Secondary school not completed* | 23 (4) | 306 (53) | 2 (0) |
| *≥ Secondary school completed* | 12 (2) | 169 (30) | 8 (1) |
| Marital Status, n (%) | | | |
| *Single* | 32 (6) | 434 (75) | 9 (2) |
| *Cohabiting or married* | 5 (1) | 77 (13) | 1 (0) |
| *Separated or divorced* | 1 (0) | 17 (3) | 0 (0) |
| Monthly income in metical's, n (%) | | | |
| *≤ 5000* | 32 (6) | 459 (79) | 10 (2) |
| *>5000* | 6 (1) | 70 (12) | 0 (0) |
| Main occupation, n (%) | | | |
| *Student* | 22 (4) | 292 (50) | 7 (1) |
| *Occasional Work* | 4 (1) | 26 (5) | 0 (0) |
| *Permanent work* | 10 (2) | 146 (25) | 0 (0) |
| *N/A* [a] | 68 (12) | | |
| Ever received blood transfusion, n (%) | | | |
| *No* | 35 (6) | 506 (88) | 9 (2) |
| *Yes* | 1 (0) | 20 (3) | 1 (0) |
| *N/A* [a] | 5 (1) | | |
| Desire to get pregnant in the next year (*n = 275 females only*), n (%) | | | |
| *No* | 8 (3) | 227 (83) | 2 (1) |
| *Yes* | 4 (1) | 30 (11) | 2 (1) |
| *N/A* [a] | 2 (1) | | |
| Age (years) of first sexual intercourse, n (%) | | | |
| *< 15 years old* | 10 (2) | 81 (14) | 1 (0) |
| *15–18 years old* | 20 (3) | 368 (64) | 5 (1) |
| *≥ 18 years* | 6 (1) | 49 (8) | 3 (0) |
| *N/A* [a] | 34 (6) | | |
| Perceived risk for HIV infection, n (%) [b] | | | |
| *No risk* | 3 (0) | 25 (4) | 0 (0) |
| *Some risk* | 32 (6) | 459 (80) | 9 (2) |
| *High risk* | 0 | 38 (7) | 1 (0) |
| *N/A* [a] | 10 (2) | | |

[a] N/A—data not available. Only shown for variables ≥ 1% of data missing.

[b] Question phrased as *"How would you classify your risk for getting infected with HIV"*.

(SD ± 3.9) years, 67% of the participants did not complete the secondary level of education. Most of the participants were single (82%) and 87% earned less than 5,000 meticais (~USD147) [24] per month, which is around the minimal national wage salary. Only 10% of the participants reported being older than 18 years of age when they had their first sexual intercourse, and almost all (94%) acknowledged that they had at least some risk of acquiring HIV. Of the 577 eligible participants who were also without HIV, 505 participants enrolled and of these 75 (15%) did not respond to the willingness to participate questionnaire at the exit visit. Independence test between levels of education revealed that participants with only primary school completed (p<0.001) were more likely to drop-out before the exit visit. The exit visit was completed by 430 participants who remained at risk for HIV.

## WTP in a HIV vaccine study

Of the 577 HIV negative participants eligible at screening, 529 (92%) said they would be willing to participate in an HIV vaccine study, 38 (7%) were not willing and 10 (2%) indicated that they did not know (Fig 1). Participants who were not willing to participate, were single 32 (6%), reported a monthly income of less than 5,000 meticais 35 (6%) (Table 2). At the exit visit, 430 (75%) participants were still active in the study and willingness to participate in a HIV vaccine study remained high, with 378 (88%) willing to participate in an HIV vaccine trial, 35 (8%) not willing, and 17 (4%) who said that they didn't know (Fig 1). Among the 430 participants who stayed throughout the duration of the study, 392 indicated willingness to participate in a vaccine trial at entry. Of these 392, 360 (92%) maintained their willingness, 19 (5%)

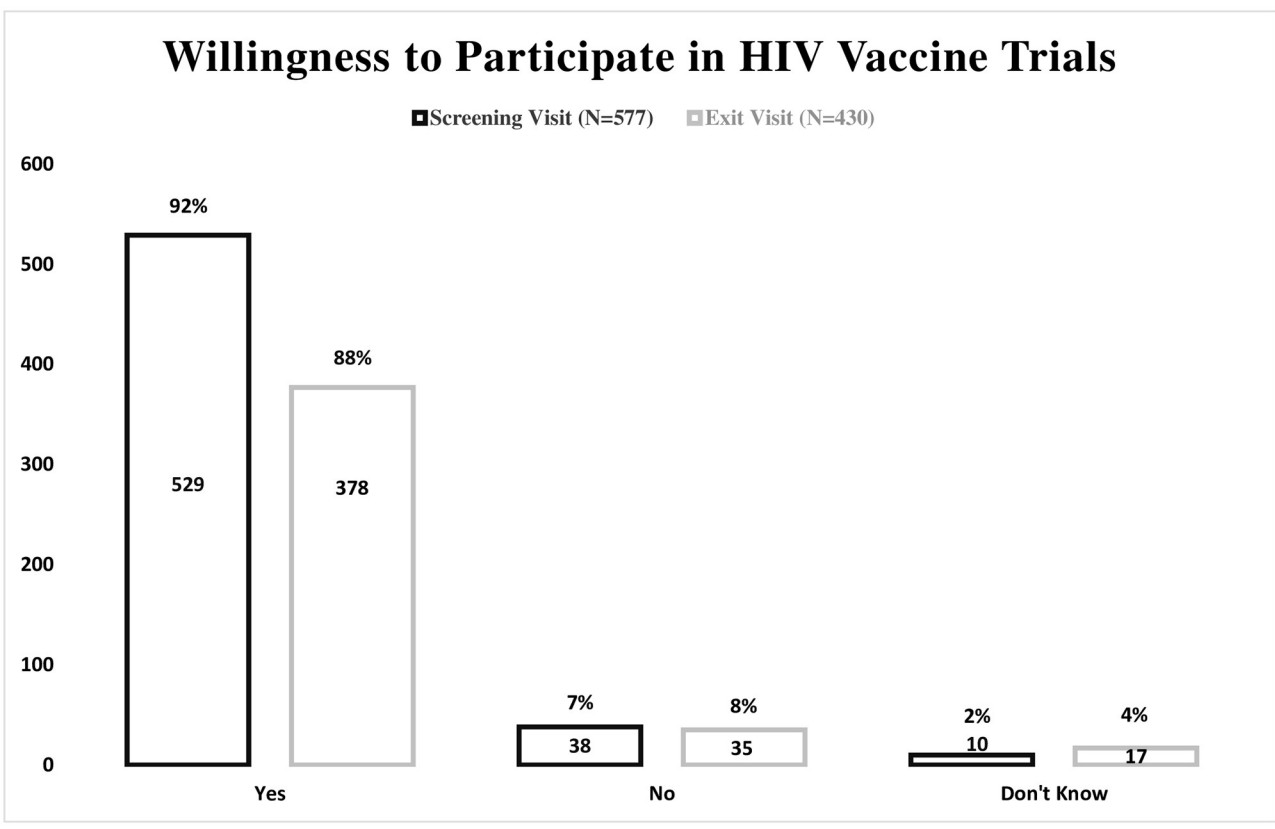

**Fig 1. Willingness to participate of 577 HIV negative participants at baseline, and of 430 HIV negative participants who completed the 24 months follow-up period.**

**Table 2. Association analysis between HIV vaccine knowledge, benefits, barriers, facilitators and motives with willingness to participate in HIV vaccine studies among the RV363 participants, by visit status.**

| Willingness to Participate Factors | Total Eligible N = 577 | Visit Status | | p-value[a] |
|---|---|---|---|---|
| | | Screening Visit (n = 430) | Exit Visit (n = 430) | |
| Purpose of a Vaccine is to prevent illness—n/N (%) | 546 (95) * | 407(95) * | 417 (97) | 0.090 |
| Had any previous information on HIV vaccine research, prior to participation—n/N (%) | 223 (39) | 175 (41) | | |
| Source of Previous information regarding HIV vaccine research—n/N (%) | | | | |
| Hospital/clinic/health worker | 43 (7) | 35 (8) | 30 (7) | 0.398 |
| Radio/Tv | 143 (25) | 112 (26) | 113 (26) | 0.917 |
| Friend/relative | 71 (12) | 56 (13) | 42 (10) | 0.016 |
| Internet | 12 (2) | 10 (2) | 30 (7) | <0.001 |
| Poster | 4 (1) | 2 (0) | 22 (5) | <0.001 |
| Learned from a vaccine trial volunteer | 3 (1) | 3 (1) | 4 (1) | 0.706 |
| Research center | 7 (1) | 7 (2) | 24 (6) | 0.002 |
| Personal benefits of participating in Vaccine study- n/N (%) | | | | |
| Learn how to avoid risky behavior | 529 (92) | 398 (93) | 352 (82) | <0.001 |
| Feel protected from HIV infection | 531 (92) | 394 (92) | 372 (87) | 0.002 |
| Get free HIV counselling and testing | 496 (86) | 369 (86) | 273 (63) | <0.001 |
| Receive updated information about HIV/AIDS | 368 (64) | 276 (64) | 253 (59) | 0.045 |
| Get small reimbursement | 83 (14) | 67 (16) | 18 (4) | <0.001 |
| Be tested for sexually transmitted infection | 527 (91) | 389 (90) | 311 (72) | <0.001 |
| Receive regular health care related to research | 373 (65) | 289 (67) | 206 (48) | <0.001 |
| Get free pregnancy test every 3 months (N = 275 females) | 195 (71) | 137 (50) | 63 (23) | <0.001 |
| Barriers that could make you not participate in a vaccine study- n/N (%) | | | | |
| Fear of needle | 26 (5) | 22 (5) | 23 (5) | 0.847 |
| Fear of getting HIV | 1 (0) | 1 (0) | 5 (1) | 0.103 |
| Fear of side effects | 15 (3) | 10 (2) | 7 (2) | 0.405 |
| Fear of death | 1 (0) | 1 (0) | 2 (0) | 0.564 |
| Fear of fetal abnormalities | 10 (2) | 9 (2) | 7 (2) | 0.480 |
| Time required for visits | 1 (0) | 1 (0) | 8 (2) | 0.008 |
| Fear of testing HIV positive | 1 (0) | 1 (0) | 4 (1) | 0.180 |
| Fear of discrimination | 2 (0) | 2 (0) | 1 (0) | 0.564 |
| How likely would you enroll in research study of a new experimental HIV vaccine if (likely[b])—n/N (%) | | | | |
| Required to come every 3 months to study center on a specific day | 571 (99) | 425 (99) | 427 (99) | 1.0000 |
| Required to talk about study experiences with a nurse | 571 (99) | 425 (99) | 427 (99) | 1.0000 |
| Required to test for HIV every 3 months | 569 (99) | 423 (98) | 427 (99) | 0.317 |
| Required to be injected with a HIV study vaccine a few times | 541 (94) | 401 (93) | 422 (98) | 0.706 |
| Required to give blood samples every time that you attend a study visit | 554 (96) | 413 (96) | 422 (98) | 0.564 |
| The center was open for weekend visits | 552 (96) | 409 (95) | 422 (98) | 0.103 |
| Required to use contraceptive during study participation (N = 275 females) | 214 (78) | 153 (56) | 172 (63) | 0.178 |
| Which factors are important for your decision to participate in a study of an experimental HIV vaccine (motives) (important[c])—n/N (%) | | | | |
| Helping vaccine research | 548 (95) * | 405 (94) * | 429 (100) | <0.001 |
| Helping fighting HIV/AIDS | 549 (95) * | 408 (95) * | 429 (100) | 0.003 |
| Positive family support | 463 (80) * | 343 (80) * | 426 (99) | <0.001 |
| Meet new people | 488 (85) * | 359 (83) * | 426 (99) | <0.001 |
| Helping the community | 553 (96) * | 408 (95) * | 429 (100) | 0.002 |
| Helping the country | 556 (96) * | 412 (96) * | 430 (100) | 0.002 |
| Helping research of HIV Vaccine | 554 (96) * | 411 (96) * | 429 (100) | 0.007 |

*(Continued)*

**Table 2.** (Continued)

| Willingness to Participate Factors | Total Eligible N = 577 | Visit Status | | p-value[a] |
|---|---|---|---|---|
| | | Screening Visit (n = 430) | Exit Visit (n = 430) | |
| *Perceived as HIV positive* | 16 (3) * | 11 (3) * | 237 (55) | <0.001 |
| *Perceive as High-risk person for HIV infection* | 9 (2) * | 7 (2) * | 233 (54) | <0.001 |
| *Unable to have sex intercourse* | 19 (3) * | 11 (3) * | 231 (54) | <0.001 |
| *People refusing contact* | 16 (3) * | 10 (2) * | 227 (53) | <0.001 |
| Preventive vaccine will work—n/N (%) | 573 (99) * | 426 (99) * | 428 (100) | 1.0000 |

[a]. McNemar's exact test.

n/N = numbers of participants who answered yes, likely or important/total numbers of participants who answered.

[b]. Likely = composite of somewhat likely and very likely.

[c]. Important = composite of important and very important.

* Data missing.

became not willing, and 13 (3%) became undecided. Of the 28 participants who initially were not willing to participate, 14 (50%) maintained their unwillingness, 12 (43%) became willing, and 2 (7%) became undecided. For 10 participants who initially stated that they did not know if they would participate in a HIV vaccine trial, 6 became willing, 2 became unwilling and 2 stayed undecided.

## HIV vaccine knowledge

At baseline, 546 (95%) participants indicated that the purpose of a vaccine is to prevent illness and 223 (39%) indicated that they had received any information on HIV vaccine research

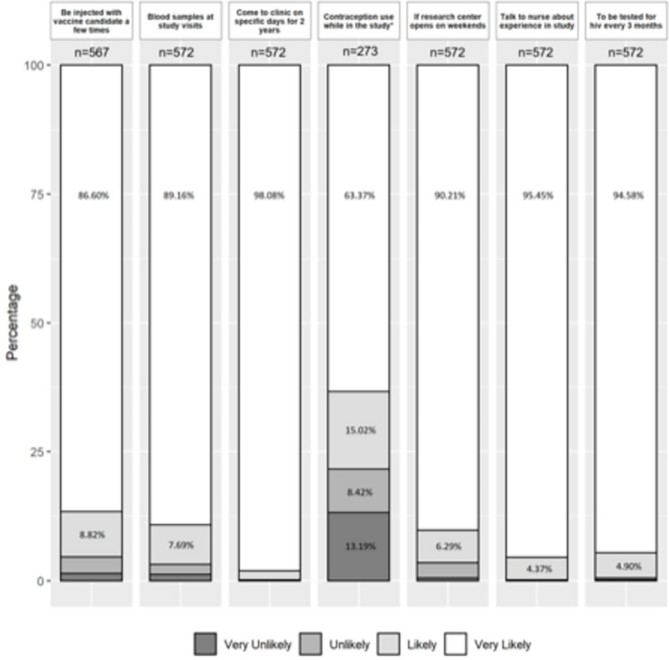

**Fig 2. Self-reported likelihood of enrolling in a research study for an experimental HIV vaccine, by theoretical study design elements.** Asterisk (*) indicates for only female. Due to bar size, percentage values of less than 4% were not shown; only subjects who provided a valid response to the question were included.

prior to study participation. The main source of information was radio/television in (n = 143, 25%), followed by friends (n = 71, 12%) and hospital (n = 43, 7%) (Table 2). Friends/relatives became a less important source of information (n = 56, 13%) vs. n = 42, 10%), (p = 0.016) at the end of the study, in contrast the internet become a more significant source of information (n = 12, 2% vs. n = 30, 7%), (p = 0.001).

## Factors that could influence HIV vaccine trial participation

Table 2 also shows the perceived personal benefits and motives that could influence decision to participate in an HIV vaccine trial for the 577 participants at screening. Feeling protected from HIV infection (n = 531, 92%), learning how to avoid HIV infection (n = 529, 92%), and free testing for sexually transmitted infections (n = 527, 91%) were the three most commonly reported personal benefits of vaccine study participation. Positive social motives for participants' decision to participate in a HIV vaccine study were helping the country (n = 556, 96%), helping the research of HIV Vaccine (n = 554, 96%), and helping the community (n = 553, 96%). Negative social motives were unable to have sex intercourse (n = 19, 3%), perceived as HIV positive (n = 16, 3%) and people refusing sexual contact (n = 16, 3%). Perceived barriers to participation in an HIV vaccine study included fear of needles (n = 26, 5%), fear of side effects (n = 15, 3%) and fear of fetal abnormalities (n = 10, 2%). When comparing data from enrollment to the 24-month visit among participants who provided information at both time-points, the proportion of perceived personal benefits of study participation tended to decrease over time while the proportion of both positive and negative social motives for study participation increased (Table 2). No statistically significant differences over time were observed for barriers to participation. When presented with various theoretical study design elements, most participants self-reported being *Very Likely* to enroll in a research study for a new experimental HIV vaccine (Fig 2).

## Predictors of willingness to participate in HIV vaccine trial

Bivariate analysis among the participants at screening visit revealed that participants who had their sexual debut between the ages of 15–18 had an odds ratio (OR) of 2.27 (95% CI: 1.02–5.04) times more likely to be willing to participate in a HIV vaccine study when compared to those whose sexual debut was before age 15 (Table 3). Perceived personal benefits of participating in an HIV vaccine study that were significantly associated in analysis with WTP include learning how to avoid risky behavior (OR 9.37, 95% CI:4.42–19.89), and feeling protected from HIV infection (OR 7.17, 95% CI:3.31–15.50). We also found that HIV vaccine trial attributes, such as having to receive an experimental HIV vaccine (OR 14.84, 95% CI: 6.02–36.6), and being required to donate blood samples (OR 7.69, 95% CI: 2.53–23.35) were associated with WTP. Multivariate analysis was adjusted for variables with p<0.05 and age as controller (Table 3). Factors identified as significant in the bivariate analysis, such as aged of sexual debut being between the ages 15–18 (aOR 2.39, 95% CI: 1.01–5.66) wanting to learn how to avoid risk behavior (aOR 4.28, 95% CI: 1.40–13.07), and feeling protected from infection (aOR 3.41, 95% CI: 1.27–9.20), were also significant in the multivariate analysis (Table 3).

Among the 430 participants who stayed through the course of the study, results from the GEE binary logistic regression (screening visit and exit visit) showed that wanting to learn how to avoid risk behaviors (aOR 3.33, 95% CI: 1.61–6.86) and feeling protected against HIV infection (aOR 2.24, 95% CI: 1.07–4.7) were associated with willingness to participate in HIV vaccine studies. The perceived risk for HIV infection (aOR 1.90, 95% CI: 0.99–3.63) shows positive association, although remaining borderline significant (Table 4).

**Table 3. Binary and multivariate logistic regression analysis of selected factors associated with willingness to participate in HIV vaccine studies, among participants of the RV363 study, at screening visit.**

| Variables | Willingness to participate | | | |
|---|---|---|---|---|
| | Screening visit assessment (n = 567)[a] | | | |
| | Bivariate Analysis | | Multivariate Analysis[b] | |
| | OR (CI 95%) | p-value | aOR (CI 95%) | p-value |
| Age (ref: *18–20 years*) | | | | |
| *21–24 years* | 0.85 (0.39–1.83) | 0.675 | 0.93 (0.39–2.23) | 0.876 |
| *25–35 years* | 0.70 (0.30–1.63) | 0.412 | 0.95 (0.36–2.52) | 0.921 |
| Gender (ref: *male*) | | | | |
| *Female* | 1.45 (0.74–2.84) | 0.280 | | |
| Education (ref: ≤ *Primary school completed*) | | | | |
| *Secondary school not completed* | 0.74 (0.21–2.55) | 0.632 | | |
| *≥ Secondary school completed* | 0.78 (0.21–2.88) | 0.712 | | |
| Age of first sexual intercourse *(ref: < 15 years old)* | | | | |
| *15–18 years old* | 2.27 (1.02–5.04) | 0.043 | 2.39 (1.01–5.66) | 0.046 |
| *≥ 18 years* | 1.01 (0.35–2.95) | 0.988 | 0.87 (0.27–2.84) | 0.819 |
| Perceived Risk for HIV infection *(ref: No risk)* | | | | |
| *Some risk* | 1.72 (0.49–6.01) | 0.395 | | |
| Previous to this study did you received information regarding HIV Vaccine research (ref: *No*) | | | | |
| *Yes* | 0.86 (0.44–1.68) | 0.662 | | |
| Which of these sources have you received information about HIV vaccine research? (ref: *None*) | | | | |
| *Hospital/clinic/health worker* | 0.66 (0.22–1.95) | 0.450 | | |
| *Radio/Tv* | 0.80 (0.39–1.66) | 0.548 | | |
| *Friend/relative* | 1.21 (0.42–3.52) | 0.724 | | |
| *Research center* | 0.42 (0.05–3.62) | 0.433 | | |
| Benefits of participating in Vaccine study (ref: *No*) | | | | |
| *Learn how to avoid risky behavior* | 9.37 (4.42–19.89) | <0.001 | 4.28 (1.40–13.07) | 0.011 |
| *Feel protected from HIV infection* | 7.17 (3.31–15.51) | <0.001 | 3.41 (1.27–9.20) | 0.015 |
| *Get free HIV counselling and testing* | 2.81 (1.33–5.93) | 0.007 | 1.00 (0.35–2.89) | 1.00 |
| *Receive updated information about HIV/AIDS* | 1.69 (0.87–3.27) | 0.121 | | |
| *Get small reimbursement* | 1.43 (0.49–4.13) | 0.513 | | |
| *Be tested for sexually transmitted infection* | 5.26 (2.43–11.42) | <0.001 | 1.48 (0.47–4.70) | 0.503 |
| *Receive regular health care related to research* | 1.76 (0.91–3.41) | 0.094 | 1.29 (0.60–2.76) | 0.513 |
| How important are the following factors in making a decision about participating in a research study of an experimental vaccine? (ref: *Not Important*) | | | | |
| *Helping vaccine research* | 0.82 (0.11–6.31) | 0.848 | | |
| *Receiving support from family and friends* | 1.17 (0.49–2.76) | 0.726 | | |
| *Meet new people* | 1.34 (0.54–3.36) | 0.530 | | |
| *Helping the community* | 1.00 (0.13–7.78) | 1.000 | | |
| *Perceived as HIV positive* | 0.94 (0.12–7.31) | 0.949 | | |
| *Perceive as High-risk person for HIV infection* | 0.43 (0.05–3.6) | 0.436 | | |

[a] 10 participants who answered "Don't know" were not included in analysis.

[b] The multivariable analysis is a complete-case analysis, n = 534, and adjusted for all significant variables with p<0.05 and age as a controller.

OR Odds Ratio.

CI Confidence Interval.

aOR Adjusted Odds Ratio.

**Table 4. Multivariate analysis using generalized estimation equation of factors associated with willingness to participate in HIV vaccine studies, among participants who complete the RV363 study (screening visit and exit visit).**

| Variables | GEE Analysis | |
|---|---|---|
| | **aOR (CI 95%)** | **p-value[c)]** |
| Age (categorized ref: *18–20 years*) | | |
| *21–24 years* | 0.88 (0.41–1.89) | 0.743 |
| *25–25 years* | 0.77 (0.34–1.71) | 0.515 |
| Gender (ref: *male*) | | |
| *Female* | 0.81 (0.42–1.58) | 0.545 |
| Age of first sexual intercourse *(ref: < 15 years old)* | | |
| *15–18 years old* | 1.76 (0.77–4.05) | 0.181 |
| *≥ 18 years* | 1.32 (0.41–4.21) | 0.643 |
| *Perceived Risk for HIV infection* (ref: No *risk*) | | |
| *Some risk* | 1.90 (0.99–3.63) | 0.053 |
| Which of these sources have you received information about HIV vaccine research? | | |
| *Friend/relative* | 1.61 (0.4–6.44) | 0.503 |
| Benefits of participating in Vaccine study (ref: *No*) | | |
| *Learn how to avoid risky behavior* | 3.33 (1.61–6.86) | <0.001 |
| *Feel protected from HIV infection* | 2.24 (1.07–4.7) | 0.032 |
| *Get free HIV counselling and testing* | 0.62 (0.27–1.44) | 0.267 |
| How important are the following factors in making a decision about participating in a research study of an experimental vaccine? (ref: *Not Important*) | | |
| *Helping the community* | 2.62 (0.55–12.42) | 0.225 |
| *Perceive as High-risk person for HIV infection* | 0.34 (0.06–2.01) | 0.236 |
| *Unable to have sex intercourse* | 3.12 (0.76–12.85) | 0.115 |
| *Not being able to contact people* | 2.17 (0.63–7.42) | 0.217 |

a) Adjusted for age; gender; age of first sexual intercourse; perceived risk for HIV infection; which of these sources have you received information about HIV vaccine research?; benefits of participating in vaccine study; How important are the following factors in making a decision about participating in a research study of an experimental vaccine?

GEE Generalized estimation equation for binary logistic regression for screening and exit visit.

aOR Adjusted Odds Ratio.

CI Confidence Interval.

## Discussion

Hypothetical willingness to participate in HIV vaccine studies was high among young adults at risk for HIV in a low-income peri-urban area in Maputo, Mozambique. This study reinforces the practicality of conducting HIV vaccine trials in Maputo. Also, we identified factors that are associated with willingness to participate over a period of time, with the same participants, which can be used for the development of communication and retention strategies of future HIV vaccine clinical trials to be conducted in Mozambique. In an HIV-incidence study cohort study with fishing communities, conducted in Uganda [25], 99.4% (n = 328) of the participants expressed their willingness to participate in a HIV vaccine trial. In contrast only 50.6% (n = 450) of young adults recruited at a youth clinic in Tanzania expressed their willingness to participate in a HIV vaccine trial [22]. These results are consistent with findings from Inugu et al. [16], where the authors concluded that willingness to participate in HIV vaccine trials varied by region and population.

Young adults and women tend to have a high incidence of HIV, and play an important role in the ongoing transmission of HIV [26,27]. This strengthens the need for HIV vaccine trials and willingness to participate studies to target the largest number of women and young volunteers [28–30]. Our population consisted of young adults most-at-risk for HIV infection, of whom almost half were women, identified and recruited from the general community, in contrast to other HIV vaccine preparedness studies which specifically targeted populations that are vulnerable and socially stigmatized such as MSM and FSW [17,22,23,31]. We found high rates of willingness to participate results for both males and females, while Meque, et al. [32] in a study conducted in another province in Mozambique, found that WTP was 77.8% (n = 1019) for females and 57.6% (n = 97) for males. In our study, the majority of our participants (both male and female) acknowledge at least some risk for HIV acquisition, which corroborates findings from other studies that perceived risk for HIV infection is related to willingness to participate [23,33,34]. Successful execution of an HIV vaccine efficacy trial requires recruitment and retention of a large number of individuals without HIV from a source population with a high incidence of disease [10], who recognize their own risk behaviors for HIV acquisition.

Altruism was the most frequently reported positive social motivator for WTP in our study, and much of the same results were reported in other studies [32,35,36], suggesting that study participation may be seen by participants as their contribution to the fight against HIV. Important to highlight, was the increase in the proportion of negative social motives. Being perceived as HIV positive was reported in a study conducted in the USA [23] and perceive as a high-risk person for HIV infection was reported in a study conducted in India [37]. In Tanzania [38] a major barrier to decline participation in phase I and phase II HIV vaccine trials was resistance from significant others, which could be related to being unable to have sex intercourse and/or people refusing contact. The ongoing presence of these factors in different contexts implies that it is necessary to define and implement recruitment strategies that address them, in order to minimize their impact. It should be noted that these variables did not rise to statistical significance in the multivariate or longitudinal modeling.

Perceived personal benefits of participating in HIV vaccine trial, such as feeling protected from HIV infection, learning how to avoid HIV infection, and free testing for HIV and STI have also been reported elsewhere [36]. The proportion of sources of information changed significantly from screening to exit visit, as information from friends and relatives dropped, but internet, poster and research center information increased. One hypothesis could be that, because of their participation in the RV363 study, participants learned how to use more accurate sources of information and, consequently, to discern what they thought was the more accurate information related to HIV vaccine research. The most common personal barrier was fear of needles, similar to other reported results [33]. We did not find fear of fetal abnormalities in other reported results. This is important, as it can impede young people who intend to have children from participating.

Willingness to participate remained high among the participants who completed the 24-month follow-up, but no assurance should be taken that actual WTP will be high. In a 2-year follow-up cohort study conducted in Kenya, Nyasani et al. [21] reported that 86 of 100 participants who expressed their willingness to participate in a HIV vaccine trial and were contacted afterwards, only 30% (n = 26) actually consented to participate in a phase-I HIV vaccine trial. As in our study, they also did not provide cohort participants with HIV vaccine research information. Asiki et al. [25], found that willingness to participate among high-risk men and women in Uganda decreased after the participants received information regarding HIV vaccine trials attributes. The principal barriers were number and duration of visits, being injected with an experimental vaccine, being required to delay pregnancy, and blood collection [21,25]. In our study, feeling protected against HIV infection, was associated with greater willingness,

highlighting the need to consider our high rate of willingness to participate with caution. Paramount to highlight, is the erroneous perception that receiving an experimental HIV vaccine could provide protection against HIV, which can lead to alterations in risk behavior due to trial participation. This can be explained by the fact that no information was given on HIV vaccine research and its goal to prove the efficacy of experimental vaccines. Equally important was the fact that the rate of perceived benefits associated with willingness to participate decreased at the end of the study, despite wanting to learn how to avoid risky behavior being associated with greater willingness to participate. We theorize that this could be for several reasons: 1) participants felt that there was not more to learn, since they received counselling for HIV risk behavior reduction during the cohort study and 2) participants might have learned more about HIV vaccine trials and, especially the fact that there is no guarantee of protection from an experimental vaccine.

Although willingness to participate (hypothetical) was high and remained high, one-quarter of the participants did not complete the willingness to participate questionnaire at the final visit. This group had the highest proportion of individuals with the lowest levels of education. We postulate that these individuals who did not respond to the willingness to participate questionnaire at the final visit can reflect the rates and characteristics of drop-out individuals who are eligible for an actual HIV vaccine trial, making it an important group to characterize in order to identify potential factors associated with HIV trial non-adherence or hesitancy. In an HIV incidence study conducted in Kenya, Nyambura et al [39] found that being female and having a higher level of education was associated with greater risk for missing a study visit.

## Limitations

There are a number of limitations to this study that should be considered. The findings are potentially limited by the high loss to follow-up from the cohort, which may lead to an overestimation of the rate of willingness to participate. This risks introduction of survivor bias into the analysis of our covariate associations, in terms of characterizing the overall cohort; however, to the extent that the intent is to characterize associations with willingness among participants who were able to complete the trial we argue that survivor bias is negligible.

Some aspects of the study may limit the generalizability of the results. Because the cohort participants were recruited based on their propensity towards high risk behaviors, they may not represent attitudes and values of a wider, lower risk or mixed-risk population. Additionally, due to the status of the CISPOC area as a high incidence population, numerous education, outreach, and testing interventions may disproportionately increase HIV awareness and willingness compared to what may be expected in areas with less outreach activity. However, within the context of high-risk behavior populations, which are those of most interest to HIV vaccine trials, we argue the results are likely generalizable. The corroboration in associations with willingness to participate in this study and the existing literature lends face validity to the generalizability of results; additionally, unlike other studies in the existing literature which limit their cohorts to small sub-populations (such as hospital-based recruitment, MSM or Trans only studies) the population-based nature of our cohort adds to the generalizability of our findings.

Participant recall bias may be an important mediator regarding reporting of previous information sources. As no information was collected on the length of time since exposure to the information source, participants who were presented with vaccine knowledge prior to screening, particularly from interpersonal sources, may not recall the incident at exit some two years later. Future studies may consider collecting time-relevant information to help address issues in recall bias as well as message permanency.

Lastly, there is a potential that social desirability bias may have influenced individual participant responses. While efforts were made to limit this tendency in participants through assurances of confidentiality of responses, neutral and impartial wording or phrasing, the potential remains that participant responses may have been influenced by the rapport developed over the cohort trial period. While it is unlikely this played a part in the knowledge-based responses (as these are more factual in nature), the potential influence of this bias should be considered when interpreting the social/personal benefits and barriers to results.

## Conclusion

A large proportion of at-risk young adults in Maputo City have expressed and maintained their WTP in future trials and should therefore be considered as participants in future trials. Young age and altruism are factors associated with willingness. However, there is a need to deepen knowledge about the associated factors, including the experimental HIV vaccine and its potential risks, to prepare better messages in order to avoid high rates of refusal during actual clinical trial recruitment.

## Supporting information

**S1 Table. The RV-363 HIV/AIDS awareness, vaccine knowledge and willingness to participate questionnaire.**
(PDF)

## Acknowledgments

The authors are thankful to the study participants who made this work possible. Institutions that supported this work included Instituto Nacional de Saúde (INS), Polana Caniço Health Research and Training Center (CISPOC), Henry M. Jackson Foundation for the Advancement of Military Medicine (HJF), HJF Medical Research International (HJFMRI), US Military HIV Research Program (MHRP), Walter Reed Army Institute of Research (WRAIR). Led by Christina Polyak, the RV363 Study Group includes: Ilesh V. Jani, Ivalda Macicame, Nilesh Bhatt, Edna Viegas, Patrícia Ramgi, Igor Capitine, Merlin L. Robb, Leigh Anne Eller, Julie Ake, Trevor Crowell, Edith M. Swann, Michelle Imbach, Steve Schech, Chiaka Nwoga, Zebiba Hassen, Qun Li, Mark Milazzo, Gail Smith, Adam Yates, Raquel Matavele Chissumba, Vanessa Monteiro, Khelvon Leonel Jose De Araujo, Carmélia Massingue, Luis Inhambizo, Ferrão Mandlate.

## Author Contributions

**Conceptualization:** Igor P. U. Capitine.

**Formal analysis:** Igor P. U. Capitine, Artur M. Uanela.

**Investigation:** Ivalda B. Macicame, Adam Yates, Mark Milazzo, Chiaka Nwoga, Ilesh V. Jani.

**Methodology:** Nilesh B. Bhatt, Nelson L. Michael, Merlin L. Robb, Ilesh V. Jani, Christina S. Polyak.

**Project administration:** Ivalda B. Macicame, Nilesh B. Bhatt, Christina S. Polyak.

**Supervision:** Nilesh B. Bhatt, Ilesh V. Jani, Christina S. Polyak.

**Validation:** Ivalda B. Macicame, Artur M. Uanela, Nilesh B. Bhatt, Adam Yates, Mark Milazzo, Chiaka Nwoga, Trevor A. Crowell, Nelson L. Michael, Merlin L. Robb, Ilesh V. Jani, Arne Kroidl, Christina S. Polyak, Caroline De Schacht.

**Writing – original draft:** Igor P. U. Capitine.

**Writing – review & editing:** Ivalda B. Macicame, Artur M. Uanela, Nilesh B. Bhatt, Adam Yates, Mark Milazzo, Chiaka Nwoga, Trevor A. Crowell, Nelson L. Michael, Merlin L. Robb, Ilesh V. Jani, Arne Kroidl, Christina S. Polyak, Caroline De Schacht.

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
