## [Decision Letter · Decision Letter 0]

6 Nov 2020

PONE-D-20-31594

Young at risk-people in Maputo City, Mozambique, present a high willingness to participate in HIV trials: results from an HIV vaccine preparedness cohort study

PLOS ONE

Dear Dr. Capitine, 

Thank you for submitting your manuscript to PLOS ONE. After careful consideration, we feel that it has merit but does not fully meet PLOS ONE’s publication criteria as it currently stands. Therefore, we invite you to submit a revised version of the manuscript that addresses the points raised during the review process.

Please kindly submit your revised manuscript by December 7, 2020 if possible. If you will need more time than this to complete your revisions, please reply to this message or contact the journal office at plosone@plos.org. Please include the following items when submitting your revised manuscript:

We look forward to receiving your revised manuscript.

Kind regards,

Karine Dubé, DrPH

Academic Editor

PLOS ONE

Journal Requirements:

2. Please provide a sample size and power calculation in the Methods, or discuss the reasons for not performing one before study initiation.

4. Please note that PLOS does not permit references to “data not shown.” Authors should provide the relevant data within the manuscript, the Supporting Information files, or in a public repository. If the data are not a core part of the research study being presented, we ask that authors remove any references to these data.

5.We note that you have indicated that data from this study are available upon request. PLOS only allows data to be available upon request if there are legal or ethical restrictions on sharing data publicly. For more information on unacceptable data access restrictions, please see http://journals.plos.org/plosone/s/data-availability#loc-unacceptable-data-access-restrictions.

6. One of the noted authors is a group or consortium [7 RV363 Study Group]. In addition to naming the author group, please list the individual authors and affiliations within this group in the acknowledgments section of your manuscript. Please also indicate clearly a lead author for this group along with a contact email address.

Reviewers' comments:

Reviewer's Responses to Questions

**Comments to the Author**

1. Is the manuscript technically sound, and do the data support the conclusions?

Reviewer #1: Yes

Reviewer #2: Yes

Reviewer #3: Partly

2. Has the statistical analysis been performed appropriately and rigorously? 

Reviewer #1: Yes

Reviewer #2: Yes

Reviewer #3: Yes

3. Have the authors made all data underlying the findings in their manuscript fully available?

Reviewer #1: Yes

Reviewer #2: No

Reviewer #3: No

4. Is the manuscript presented in an intelligible fashion and written in standard English?

Reviewer #1: Yes

Reviewer #2: Yes

Reviewer #3: Yes

5. Review Comments to the Author

Reviewer #1: Kudos for your success in enrolling and following this sizable cohort for 24 months.

INTRODUCTION

I note your statement that the importance of WTP factors varies over time and contexts (lines 67-68). How then should we assess the meaning of the findings of these interviews done between 2013 and 2017? Has the epidemic changed? Have attitudes changed?

The fact that 2 HIV vaccine trials have been conducted in Mozambique (including a Phase 2 trial) and more are currently under way seems like the best, most concrete demonstration of WTP (lines 57-60). Might there be information about enrollment and follow-up from those trials that would be relevant to the discussion of this WTP study?

RESULTS

The cohort comprises young people at substantial risk of HIV infection, the great majority of whom are aware of that risk. It seems like an ideal target for HIV vaccine trials. So the high WTP percentages are no surprise to me.

The low loss to follow-up (LFU) and the continued high WTP at the end of follow-up testify to a well-done study that treated the participants well!

Line 209: you state that motives for not participating increase during follow-up. I do not see the evidence for that.

Table 3: the OR for "required to receive the vaccine a few times" increased from 14.8 to 104.8 after adjustment. I have never seen an aOR that high! Can you look more deeply into the data to find out what is going on? Perhaps some major collinearity.

Table 3: The factor "required to use contraceptive" is associated with an increased WTP. In Table 2, that factor seems to reduce participants' WTP. (The Meque study also found that a family planning requirement would be a barrier for some women.) Can you examine that further? In Table 2, you show data for women only. Perhaps the results in Table 3 include the responses from both women and men?

DISCUSSION

Lines 264-265: you state that the cohort derives from the general population. I would reiterate that they are nevertheless at substantial risk of contracting HIV, as borne out by numerous prevalence and incidence estimations.

Line 282: rather than "the correct information" I would write "more accurate information."

Lines 283-285: I am not clear on your argument here. Table 2 shows that the perceived value or salience of the potential benefits of vaccine trial participation diminished during follow-up. But knowledge of the potential benefits would not go down, would it? Admittedly, in line 305 you write that no information was given to them on features of potential vaccine research. But I would say that their responses about possible personal benefits shown in Table 2 reveal a pretty clear idea of what a vaccine trial might offer.

Lines 321-325: your brief discussion of limitations made me curious whether there was a larger cohort that was created for RV 363, since you refer to the RV 363 incidence cohort in line 323. I looked at reference #19, but it describes only the cross-sectional findings from the baseline screening of 1125 participants. Was the larger cohort also followed for 24 months? If so, can you provide the LFU figure from the entire RV 363 study? That seems like a very relevant and important piece of data to help interpret these findings from a subset of participants. It could be inserted into the section Study Design and Populations, and then discussed in limitations. If only the 577 participants described here were followed prospectively, then these questions of mine are irrelevant.

Reviewer #2: General comments:

This study looks at the willingness to participate in a HIV vaccine trial among young adults at a suburban area in Maputo City, Mozambique. Authors assessed factors and barriers to WTP in a hypothetical HIV vaccine trial. This is quite an interesting manuscript, and the authors set out the aims quite clearly. However, there is a need to explore the uniqueness of the manuscript given that there is a relatively extensive literature on factors about WTP in HIV vaccine trial in different African populations as well as strengths form this study.

Some specific points need authors consideration:

Introduction

The authors mentioned that there are already studies conducted in Africa assessing factors associated with WTP, including in Mozambique; so, what is new about the current study that other studies did not assess? Authors may need to explore additional gaps to justify the need for this study.

Methods

Authors may need to provide additional description of all variables in the methods section.

Lines 120- 121: how many participants answering do not know, refused to answer, someone else should decided etc were removed from the analyses? This need to be specified.

Line 130: What are willingness to participate factors? Is this different from the willingness to participate (the dependent variable)? This is not clear ; How motives and barriers to WTP were assessed? This is not clear in the methods section

How the authors defined perceived risk for HIV infection? How was this explanined to the participants?

Results

Table1. How the authors define perceived risk for HIV infection? See comment above

Figure 1 is missing

Table 2: Do WTP factors the same as HIV vaccine knowledge? please see comment above on line 130

Table 3: Title … analysis of selected factors for willingness to participate… Please replace "for" by "associated" to make the title more clear.

How do the authors explain the finding of volunteers requiring to be injected with an HIV vaccine express WTP, and on the other hand, fear of needles a barrier?

Table 4: perceived risk for HIV infection was not significant. Authors may reconsider from including it as a factor associated with WTP in a vaccine trial

Discussion:

Lines 261-262- authors mentioned the need for WTP studies to target women… and line 267 – authors found high rates of WTP for men and women. Did the authors assess gender differences in the WTP in the current study? How WTP differs from men and women in the current study at screening and exit visits? Given that this study has a gender balance in participation authors may want to assess gender differences in factors and barriers to WTP in a vaccine trial.

Line 287: I am not entirely clear about this sentence, as the authors did not assess this factor in the current study. Did the authors find gender differences in assessing the barriers to participating in vaccine trials? Please see comment above.

Lines 312-314: A substantial percentage (25%) of the participants did not complete the WTP questionnaire at the exit visit. With such a considerable number of participants, the author may need to provide potential explanations and assess the characteristics of this group. How this group of participants (n=147) differ from the group who answered no to the WTP questionnaire at the exit visit? Could the authors discuss their similarities and differences?

Lines 322-324: The authors discussed the limitations of the study. Can the authors discuss the generalizability of this study? What % of participants came from different suburban areas other than Polana Caniço? This is a relevant issue in this study as if I understood the majority o participants are from Polana Caniço area.

Along these lines, it is also essential to discuss the strengths of the study as this can add more relevance to the study.

Reviewer #3: General Comments:

This study assessed factors associated with willingness to participate in potential HIV vaccine trials by conducting a 24-month observational cohort study among high-risk people in Maputo City, Mozambique. The analyses presented make use of data collected during a 24-month prospective HIV incidence study. As currently formulated, the analyses do not provide new insights into who might be interested – and who should be the focus of – future HIV vaccine trials. As in several studies, willingness to participate is most associated with HIV risk perception and/or altruism. The study could benefit from inclusion of HIV incidence (which I believe was the main outcome of the parent study) in models of WTP. In addition, some of the measures used in this study likely suffer from self-presentation bias. It might be more informative to compare likert scale response options (“very” likely versus other categories) than to simply collapse into bivariate variables. Finally, the authors should provide a more thorough discussion in the introduction and discussion sections about differences between “willingness” and actual behavior.

Introduction:

1) The authors should consider how to strengthen the introduction and justification for the research presented in this manuscript. Specifically, the introduction should address the idea that hypothetical “willingness” may not correlate with actual behavior and, to the extent possible, identify any factors that do strongly correlate between the two. Given that barriers and facilitators of trial participation vary widely over time, population and context, what can we learn from these kind of studies?

Methods:

2) Based on description of this study, participants enrolled in an HIV incidence study that did not entail receiving any kind of “vaccine-like intervention”. In order to interpret findings related to “willingness”, it would be useful to provide the actual script that described what participation in an HIV vaccine trial would entail.

3) What proportion of participants who joined the HIV incidence study responded to questions about willingness to participate in future HIV vaccine trials? (It is unclear whether the 570 of 1150 screened volunteers related only to the “willingness to participate” sample, or to the parent study in which these data were collected.

4) Were questions administered via tablet, or on paper? (I assume face-to-face interviews, correct?)

5) It looks like the main willingness question was bimodal. However, the “barriers” question with likert scale responses also seems to be to be more about willingness than barriers. If specific barriers to participation were assessed, please provide an idea about these items. The same would be useful for the “facilitators” items.

Results:

6) Given the lack of variation in the willingness to participate variable, the authors might consider focusing their analysis on the “barriers” variable instead, maintaining the 4 point likert response scale.

7) It is interesting that all perceived benefits of trial participation dropped between screening and exit. Do the authors have any ideas about why this is, or who participants were that have a decrease in perceived benefits over time?

8) Given the likelihood of self-presentation bias, it might be useful to examine and/or provide data for strongly agree category versus other categories related to willingness to participant if various tests, procedures required.

9) It appears that willingness to participate largely driven by risk perception for this study. How does this compare to other studies?

Discussion

10) The statement about the “practicality” of conducting willingness to participate studies seems overly positive. WTP studies are expensive and themselves may be considered interventions that drive incidence down – so promoting them as practical is not very convincing. It would be useful to show how such trials can help to better target the right kind of participants, recruit and retain them.

11) Line 254/page 18 – “study” redundant.

12) The discussion brings to mind that this study would be strengthened if the authors included HIV incidence as covariate in the models. Are those who are willing to participate also most likely to be at risk of HIV? It is particularly important to determine ways to recruit participants from the general population who are in fact most likely to be at risk of HIV – those who are altruistic are also likely to be at lower risk.

13) Changes in source of information – YES, of course. These studies are interventions in themselves. But, not a practical way to inform people about HIV vaccine trials.

14) Study limitations should identify self-presentation bias.

6. PLOS authors have the option to publish the peer review history of their article (what does this mean?). If published, this will include your full peer review and any attached files.

Reviewer #1: No

Reviewer #2: No

Reviewer #3: No

---

## [Author Response · Author response to Decision Letter 0]

19 Apr 2021

Editor Comments

1. Please ensure that your manuscript meets PLOS ONE's style requirements, including those for file naming. The PLOS ONE style templates can be found at.

R: Manuscript updated according to PLOS ONE’s style requirements.

2. Please provide a sample size and power calculation in the Methods or discuss the reasons for not performing one before study initiation.

R: We thank you for the comments. We added section regarding the calculation of the sample size added (Line 100-103) accordingly.

R: The questionnaire was developed by the study team. The questionnaire was not translated into Portuguese, but the study interviewers were trained to administer the questionnaire in Portuguese. Questionnaire added as S1 RV363-WTP_Questionnaire (line 133).

4. Please note that PLOS does not permit references to “data not shown.” Authors should provide the relevant data within the manuscript, the Supporting Information files, or in a public repository. If the data are not a core part of the research study being presented, we ask that authors remove any references to these data. 

R: We thank you for the comments. The information has been removed from the manuscript.

5. We note that you have indicated that data from this study are available upon request. PLOS only allows data to be available upon request if there are legal or ethical restrictions on sharing data publicly. For more information on unacceptable data access restrictions, please see: http://journals.plos.org/plosone/s/data-availability#loc-unacceptable-data-access-restrictions.

a. If there are ethical or legal restrictions on sharing a de-identified data set, please explain them in detail (e.g., data contain potentially sensitive information, data are owned by a third-party organization, etc.) and who has imposed them (e.g., an ethics committee). Please also provide contact information for a data access committee, ethics committee, or other institutional body to which data requests may be sent.

b. If there are no restrictions, please upload the minimal anonymized data set necessary to replicate your study findings as either Supporting Information files or to a stable, public repository and provide us with the relevant URLs, DOIs, or accession numbers. For a list of acceptable repositories, please see http://journals.plos.org/plosone/s/data-availability#loc-recommended-repositories.

R. Data Availability statement was updated when re-submitting. There are currently no ethical or legal restrictions on sharing the de-identified data set. The dataset supporting the conclusions of this article will be made available in the following Harvard Dataverse repository prior to publication of this article at the following link: https://doi.org/10.7910/DVN/RBTW80

6. One of the noted authors is a group or consortium [7 RV363 Study Group]. In addition to naming the author group, please list the individual authors and affiliations within this group in the acknowledgments section of your manuscript. Please also indicate clearly a lead author for this group along with a contact email address.

R: Thank you for your comment. We have included a list of individual authors and affiliations for the ‘RV363 Study Group’ to the Acknowledgements as suggested.

Reviewer#1 Comments

Kudos for your success in enrolling and following this sizable cohort for 24 months.

1. Introduction

a. I note your statement that the importance of WTP factors varies over time and contexts (lines 67-68). How then should we assess the meaning of the findings of these interviews done between 2013 and 2017? Has the epidemic changed? Have attitudes changed?

R: During the period 2013 and 2017, national reports indicate that HIV prevalence has remained stable. One of the principal findings of this study, is that the perceived personal benefits for wanting to participate in a HIV vaccine trial varied between the same period (table 2). This was also further discussed in the discussion section (line 306 – 313). 

b. The fact that 2 HIV vaccine trials have been conducted in Mozambique (including a Phase 2 trial) and more are currently under way seems like the best, most concrete demonstration of WTP (lines 57-60). Might there be information about enrollment and follow-up from those trials that would be relevant to the discussion of this WTP study?

R: We agree that the data about recruitment and follow-up of previous participants of HIV vaccine clinical trials conducted in Mozambique would be relevant to the discussion, since it would represent actual reasons for participating or not in a HIV clinical trial conducted in Mozambique. However, both trials did not report the reasons for participating or reasons to withdraw from participating in HIV vaccine study.

2. Results

a. The cohort comprises young people at substantial risk of HIV infection, the great majority of whom are aware of that risk. It seems like an ideal target for HIV vaccine trials. So, the high WTP percentages are no surprise to me.

R: Although the high rates of willingness to participate are encouraging results, until the time of implementing the study it was not possible to know whether the willingness to participate would be high or low. A study carried out in Tanzania with young people with similar characteristics reported rates of willingness to participate below compared to the results reported in this article. Literature reviews also reported that willingness to participate varies.

c. The low loss to follow-up (LFU) and the continued high WTP at the end of follow-up testify to a well-done study that treated the participants well!

R: We appreciate this compliment. As part of a team effort, it was possible to retain most of the participants.

d. Line 209: you state that motives for not participating increase during follow-up. I do not see the evidence for that. 

R: We updated the information. Instead of motives for not participating it should be important motives. We also added proportions to clarify (line 227-230).

e. Table 3: the OR for "required to receive the vaccine a few times" increased from 14.8 to 104.8 after adjustment. I have never seen an aOR that high! Can you look more deeply into the data to find out what is going on? Perhaps some major collinearity. 

R: Thank you for this very insightful comment. Upon further investigation, we found that there were some variables which were moderately correlated with the outcome, namely “required to receive the vaccine a few times” and “required to donate blood samples”, so these were removed. Additionally, the model included two factors which were specific for women (“required to use contraceptive” and “get free pregnancy test every 3 months”), therefore, these were also removed which added men back into the multivariate model in Table 3. The 4-point Likert scale data pertaining to ‘How likely would you enroll in research study of a new experimental HIV Vaccine if’ are now presented as a stacked column chart in Figure 2.

f. Table 3: The factor "required to use contraceptive" is associated with an increased WTP. In Table 2, that factor seems to reduce participants' WTP. (The Meque study also found that a family planning requirement would be a barrier for some women.) Can you examine that further? In Table 2, you show data for women only. Perhaps the results in Table 3 include the responses from both women and men?

R: Data in Table 2 represent both men and women. The question of “required to use contraceptive” as well as “get free pregnancy test every 3 months” was only answered by women. For this reason, in our updated model for bivariate and multivariate analysis we removed these factors in Table 3. Table 3 now includes responses for both men and women. 

g. Lines 264-265: you state that the cohort derives from the general population. I would reiterate that they are nevertheless at substantial risk of contracting HIV, as borne out by numerous prevalence and incidence estimations.

R: We agree with your comments. We clarified that our participants are young adults most-at-risk for HIV infection (Line 291-294).

b. Line 282: rather than "the correct information" I would write "more accurate information."

R: We thank you for the comments. Information updated in the manuscript as requested (Line 310).

c. Lines 283-285: I am not clear on your argument here. Table 2 shows that the perceived value or salience of the potential benefits of vaccine trial participation diminished during follow-up. But knowledge of the potential benefits would not go down, would it? Admittedly, in line 305 you write that no information was given to them on features of potential vaccine research. But I would say that their responses about possible personal benefits shown in Table 2 reveal a pretty clear idea of what a vaccine trial might offer.

R: In the discussion section we hypothesize that, the participants learned to obtain more accurate information from more reliable sources, simply because they participated in the study. However, a good part of the participants expected that by being injected with an experimental HIV vaccine, they could somehow get some protection against HIV infection. This phenomenon has been observed in other studies as well, as described in the discussion (Line 326 -339), and it also shows the importance of this study, because one of the recommendations is that detailed information regarding HIV vaccine trials should be clearer provided to study participants, to avoid high percentages of study withdraw (Line 325-326). 

d. Lines 321-325: your brief discussion of limitations made me curious whether there was a larger cohort that was created for RV 363, since you refer to the RV 363 incidence cohort in line 323. I looked at reference #19, but it describes only the cross-sectional findings from the baseline screening of 1125 participants. Was the larger cohort also followed for 24 months? If so, can you provide the LFU figure from the entire RV 363 study? That seems like a very relevant and important piece of data to help interpret these findings from a subset of participants. It could be inserted into the section Study Design and Populations, and then discussed in limitations. If only the 577 participants described here were followed prospectively, then these questions of mine are irrelevant. 

R: Data on the incidence study is yet to be published and will be published as a separate article. Only 577 participants were eligible and followed. 

Reviewer#2 Comments

This study looks at the willingness to participate in a HIV vaccine trial among young adults at a suburban area in Maputo City, Mozambique. Authors assessed factors and barriers to WTP in a hypothetical HIV vaccine trial. This is quite an interesting manuscript, and the authors set out the aims quite clearly. However, there is a need to explore the uniqueness of the manuscript given that there is a relatively extensive literature on factors about WTP in HIV vaccine trial in different African populations as well as strengths form this study. Some specific points need authors consideration:

1. Introduction 

a. The authors mentioned that there are already studies conducted in Africa assessing factors associated with WTP, including in Mozambique; so, what is new about the current study that other studies did not assess? Authors may need to explore additional gaps to justify the need for this study.

R: We appreciate the comments. We have edited the introduction section to make the objectives and justification for the study clearer.

2. Methods

b. Authors may need to provide additional description of all variables in the methods section.

R: We added the questionnaire, as requested (Line 133).

c. Lines 120- 121: how many participants answering do not know, refused to answer, someone else should decided etc were removed from the analyses? This need to be specified.

R: We appreciated the recommendation. Information updated in the manuscript as requested (line 140 – 142).

d. Line 130: What are willingness to participate factors? Is this different from the willingness to participate (the dependent variable)? This is not clear; How motives and barriers to WTP were assessed? This is not clear in the methods section

R: We thank you for the comments. We added information that clarifies the reasons for not being willing to participate, the benefits of being willing to participate, the barriers and facilitators to participate in an HIV vaccine study – Willingness to participate section (Line 116-133). Questionnaire also added.

e. How the authors defined perceived risk for HIV infection? How was this explained to the participants?

R: The question was: “How would you classify your risk for getting infected with HIV” – information added to Table 1 legend (Line 183).

3. Results

f. Table1. How the authors define perceived risk for HIV infection? See comment above

R: Please see response above.

g. Figure 1 is missing

R: Figure 1 was added as a separate file. We will review journal guidelines and make sure we comply with journal guidelines.

h. Table 2: Do WTP factors the same as HIV vaccine knowledge? please see comment above on line 130

R: HIV vaccine knowledge is different form WTP variable. Please see response above and refer to study questionnaire.

i. Table 3: Title … analysis of selected factors for willingness to participate… Please replace "for" by "associated" to make the title more clear.

R: Thank you for the suggested edits. We have updated the information accordingly.

j. How do the authors explain the finding of volunteers requiring to be injected with an HIV vaccine express WTP, and on the other hand, fear of needles a barrier?

R: We believe that because the participants did not receive a detailed information about HIV vaccine clinical trials attributes, they might have thought that an experimental HIV vaccine would at some point provide protection. We address this issue in the discussion section. (Line 327-339). We believe that with our results we can’t explain why fear of needles is a barrier. Other studies have also identified this as a barrier. We believe a more in-depth analysis with a qualitative study would provide the best answers. 

k. Table 4: perceived risk for HIV infection was not significant. Authors may reconsider from including it as a factor associated with WTP in a vaccine trial

R: We thank you for your comments. The Perceived risk was borderline significant p=0.053. We believe this trend should be considered.

4. Discussion

l. Lines 261-262- authors mentioned the need for WTP studies to target women… and line 267 – authors found high rates of WTP for men and women. Did the authors assess gender differences in the WTP in the current study? How WTP differs from men and women in the current study at screening and exit visits? Given that this study has a gender balance in participation authors may want to assess gender differences in factors and barriers to WTP in a vaccine trial. – 

R: We did not observe any difference in willingness to participate between male and female, but because of the high HIV infection among young and females we recognize the importance of them participating in HIV vaccine trials.

m. Line 287: I am not entirely clear about this sentence, as the authors did not assess this factor in the current study. Did the authors find gender differences in assessing the barriers to participating in vaccine trials? Please see comment above. 

R: We appreciate your comments. We have updated the information and add “young people who intend to have children from participating (Line 314-316)”.

n. Lines 312-314: A substantial percentage (25%) of the participants did not complete the WTP questionnaire at the exit visit. With such a considerable number of participants, the author may need to provide potential explanations and assess the characteristics of this group. How this group of participants (n=147) differ from the group who answered no to the WTP questionnaire at the exit visit? Could the authors discuss their similarities and differences?

R: We recognize that it is important to characterize the lost to follow-up group. However, this information will be presented and discussed in the main article on HIV incidence. The team had discussed this issue and have addressed in the discussion.

o. Lines 322-324: The authors discussed the limitations of the study. Can the authors discuss the generalizability of this study? What % of participants came from different suburban areas other than Polana Caniço? This is a relevant issue in this study as if I understood the majority o participants are from Polana Caniço area. 

R: We thank you for the comments. Data will be used in another manuscript (incidence paper).

p. Along these lines, it is also essential to discuss the strengths of the study as this can add more relevance to the study. 

R: We thank you for the suggestions. Please refer to the discussion section for an update discussion regarding the study strengths (Line 279-284).

Reviewer#3 Comments

This study assessed factors associated with willingness to participate in potential HIV vaccine trials by conducting a 24-month observational cohort study among high-risk people in Maputo City, Mozambique. The analyses presented make use of data collected during a 24-month prospective HIV incidence study. As currently formulated, the analyses do not provide new insights into who might be interested – and who should be the focus of – future HIV vaccine trials. As in several studies, willingness to participate is most associated with HIV risk perception and/or altruism. The study could benefit from inclusion of HIV incidence (which I believe was the main outcome of the parent study) in models of WTP. In addition, some of the measures used in this study likely suffer from self-presentation bias. It might be more informative to compare likert scale response options (“very” likely versus other categories) than to simply collapse into bivariate variables. Finally, the authors should provide a more thorough discussion in the introduction and discussion sections about differences between “willingness” and actual behavior.

a. Introduction - The authors should consider how to strengthen the introduction and justification for the research presented in this manuscript. Specifically, the introduction should address the idea that hypothetical “willingness” may not correlate with actual behavior and, to the extent possible, identify any factors that do strongly correlate between the two. Given that barriers and facilitators of trial participation vary widely over time, population and context, what can we learn from this kind of studies?

R: Updated the introduction section with a clearer information that justifies why this study was conducted (please refer to reviewer 2 comments response). We agree that hypothetical willingness to participate does not correlated with actual willingness to participate and address it in the discussion section (line 299 – 303).

1. Methods 

b. Based on description of this study, participants enrolled in an HIV incidence study that did not entail receiving any kind of “vaccine-like intervention”. In order to interpret findings related to “willingness”, it would be useful to provide the actual script that described what participation in an HIV vaccine trial would entail.

R: Information added to the section questionnaire. We added the questionnaire, as requested (Line 133).

c. What proportion of participants who joined the HIV incidence study responded to questions about willingness to participate in future HIV vaccine trials? (It is unclear whether the 570 of 1150 screened volunteers related only to the “willingness to participate” sample, or to the parent study in which these data were collected. 

R: Only eligible participants who answered the questionnaire were included in the analyses. In totally 1150 participants were screened, of whom 577 eligible participants answered the questionnaire. Information updated in the methods sections (line 97-99).

d. Were questions administered via tablet, or on paper? (I assume face-to-face interviews, correct?) 

R: Paper based questionnaires were administered face-to-face to the participants. Information updated (line 108 - 109).

e. It looks like the main willingness question was bimodal. However, the “barriers” question with likert scale responses also seems to be to be more about willingness than barriers. If specific barriers to participation were assessed, please provide an idea about these items. The same would be useful for the “facilitators” items.

R: Please see updated information regarding the methods and results section. Titles were updated, description of variables also. Table 2 was also updated.

2. Results

f. Given the lack of variation in the willingness to participate variable, the authors might consider focusing their analysis on the “barriers” variable instead, maintaining the 4-point likert response scale. 

R: We appreciate the suggestion. But we believe that our results respond to the objective of this article.

g. It is interesting that all perceived benefits of trial participation dropped between screening and exit. Do the authors have any ideas about why this is, or who participants were that have a decrease in perceived benefits over time? 

R: We agree that the results are interesting. We address it in the discussion section. (Line 311 – 316).

h. Given the likelihood of self-presentation bias, it might be useful to examine and/or provide data for strongly agree category versus other categories related to willingness to participant if various tests, procedures required.

R: We appreciate the suggestion. But we think it’s out the scope of this article.

i. It appears that willingness to participate largely driven by risk perception for this study. How does this compare to other studies?

R: The same results were reported in other studies. We address it in the discussion section. (Line 300 – 303).

3. Discussion

j. The statement about the “practicality” of conducting willingness to participate studies seems overly positive. WTP studies are expensive and themselves may be considered interventions that drive incidence down – so promoting them as practical is not very convincing. It would be useful to show how such trials can help to better target the right kind of participants, recruit and retain them.

R: Regarding practicality, we were referring to the practicality of conducting HIV clinical trials in Maputo (Line 280-281).

k. Line 254/page 18 – “study” redundant.

R: We appreciate the suggestions. Information updated.

l. The discussion brings to mind that this study would be strengthened if the authors included HIV incidence as covariate in the models. Are those who are willing to participate also most likely to be at risk of HIV? It is particularly important to determine ways to recruit participants from the general population who are in fact most likely to be at risk of HIV – those who are altruistic are also likely to be at lower risk. 

R: Data on the incidence study is yet to be published and will be published as a separate article.

m. Changes in source of information – YES, of course. These studies are interventions in themselves. But not a practical way to inform people about HIV vaccine trials. – 

R: We agree with your comments. However, we think that the results of this study can help inform future studies and recruitment and retention strategies, regarding what information should be provided to the participants. This study results made it clear that it is important to provide participants with clear and detailed information about HIV vaccine clinical trials before and during their participation.

n. Study limitations should identify self-presentation bias. 

R: We recognize that how the participant feels can impact how he responds, and that’s a human phenomenon. We believe that with our current results, we don’t have enough evidence to discuss it.

---

## [Decision Letter · Decision Letter 1]

27 May 2021

PONE-D-20-31594R1

Young at risk-people in Maputo City, Mozambique, present a high willingness to participate in HIV trials: results from an HIV vaccine preparedness cohort study

PLOS ONE

Dear Dr. Capitine,

Thank you for submitting your manuscript to PLOS ONE. After careful consideration, we feel that it has merit but does not fully meet PLOS ONE’s publication criteria as it currently stands. Therefore, we invite you to submit a revised version of the manuscript that addresses the points raised during the review process.

We look forward to receiving your revised manuscript.

Kind regards,

Karine Dubé, DrPH

Academic Editor

PLOS ONE

Journal Requirements:

Reviewers' comments:

Reviewer's Responses to Questions

**Comments to the Author**

1. If the authors have adequately addressed your comments raised in a previous round of review and you feel that this manuscript is now acceptable for publication, you may indicate that here to bypass the “Comments to the Author” section, enter your conflict of interest statement in the “Confidential to Editor” section, and submit your "Accept" recommendation.

Reviewer #1: All comments have been addressed

Reviewer #2: (No Response)

Reviewer #3: All comments have been addressed

2. Is the manuscript technically sound, and do the data support the conclusions?

Reviewer #1: Yes

Reviewer #2: Yes

Reviewer #3: Yes

3. Has the statistical analysis been performed appropriately and rigorously? 

Reviewer #1: Yes

Reviewer #2: I Don't Know

Reviewer #3: Yes

4. Have the authors made all data underlying the findings in their manuscript fully available?

Reviewer #1: Yes

Reviewer #2: (No Response)

Reviewer #3: Yes

5. Is the manuscript presented in an intelligible fashion and written in standard English?

Reviewer #1: Yes

Reviewer #2: (No Response)

Reviewer #3: Yes

6. Review Comments to the Author

Reviewer #1: (No Response)

Reviewer #2: I again congratulate the authors for such an interesting paper that helps to advance evidence of HIV vaccines. I also thank the authors for addressing the majority of issues in the manuscript. However, there are still some issues that need authors consideration as they are relevant. I leave to the Editor to decide the paper's outcome, to accept as it is, or to request another revision based on the reviewer's findings.

1. One issue is related to the interpretation of a borderline/ non-significant result on table 4. Significance and non-significance are based on an arbitrary cut-off of 5% and as such one needs to interpret according to the study context. Although I understand authors need report their bordeline result, I still believe authors may not report it as it is a significant finding, without a further explanation. Instead, they may need to discuss the meaning of this finding in terms of their impact or so. This may be an issue with sample size, but the meaning needs to be discussed. On page 19 (see below the text extract), the authors report the same findings as purely significant. This in my view, needs attention.

"Among the 430 participants who stayed through the course of the study, results from the GEE

265 binary logistic regression (screening visit and exit visit) showed that perceived risk for HIV

266 infection (aOR 1.90, 95% CI: 0.99-3.63),… were associated with willingness to participate in HIV vaccine studies (Table 4)."

k.Table 4: perceived risk for HIV infection was not significant. Authors may reconsider

from including it as a factor associated with WTP in a vaccine trial

R: We thank you for your comments. The Perceived risk was borderline significant

p=0.053. We believe this trend should be considered.

2. The second issue is related to the limitations of the study that needs to be clearly addressed. In my previous revision, I raised the issue of the generalizability of the study that needed to be discussed. However, the authors mentioned that this issue would be addressed in another paper, not the current one (see below the authors’ answer). I still believe that the issue of generalizability needs to be mentioned in this paper. There are other issues that were postponed to a future paper, but are also important in the current paper.

o.Lines 322-324: The authors discussed the limitations of the study. Can the authors

discuss the generalizability of this study? What % of participants came from different

suburban areas other than Polana Caniço? This is a relevant issue in this study as if I

understood the majority o participants are from Polana Caniço area.

R: We thank you for the comments. Data will be used in another manuscript (incidence

paper).

3. The third issue is acknowledging the self-presentation bias that another reviewer (#3) had raised, but was not addressed by the authors. I also state that these type of studies are highly influenced by social desirability bias—this need to be considered in the limitations and actions to minimise it stated.

Reviewer #3: (No Response)

7. PLOS authors have the option to publish the peer review history of their article (what does this mean?). If published, this will include your full peer review and any attached files.

Reviewer #1: No

Reviewer #2: No

Reviewer #3: No

---

## [Author Response · Author response to Decision Letter 1]

26 Jul 2021

R: We would like to take this opportunity to thank you for your comments and suggestions, which we believe will improve the manuscript. Please find the answers below.

1. One issue is related to the interpretation of a borderline/ non-significant result on table 4. Significance and non-significance are based on an arbitrary cut-off of 5% and as such one needs to interpret according to the study context. Although I understand authors need report their bordeline result, I still believe authors may not report it as it is a significant finding, without a further explanation. Instead, they may need to discuss the meaning of this finding in terms of their impact or so. This may be an issue with sample size, but the meaning needs to be discussed. On page 19 (see below the text extract), the authors report the same findings as purely significant. This in my view, needs attention.

"Among the 430 participants who stayed through the course of the study, results from the GEE 265 binary logistic regression (screening visit and exit visit) showed that perceived risk for HIV 266 infection (aOR 1.90, 95% CI: 0.99-3.63),… were associated with willingness to participate in HIV vaccine studies (Table 4)."

“k.Table 4: perceived risk for HIV infection was not significant. Authors may reconsider from including it as a factor associated with WTP in a vaccine trial R: We thank you for your comments. The Perceived risk was borderline significant p=0.053. We believe this trend should be considered.”

R: Thank you, this comment was well taken. We have clarified the marginally significant results by revising the text as follows:

“Among the 430 participants who stayed through the course of the study, results from the GEE binary logistic regression (screening visit and exit visit) showed that wanting to learn how to avoid risk behaviors (aOR 3.33, 95% CI: 1.61-6.86) and feeling protected against HIV infection (aOR 2.24, 95% CI: 1.07-4.7) were associated with willingness to participate in HIV vaccine studies. The perceived risk for HIV infection (aOR 1.90, 95% CI: 0.99-3.63) shows positive association, although remaining borderline significant (Table 4).”

2. The second issue is related to the limitations of the study that needs to be clearly addressed. In my previous revision, I raised the issue of the generalizability of the study that needed to be discussed. However, the authors mentioned that this issue would be addressed in another paper, not the current one (see below the authors’ answer). I still believe that the issue of generalizability needs to be mentioned in this paper. There are other issues that were postponed to a future paper, but are also important in the current paper.

“o.Lines 322-324: The authors discussed the limitations of the study. Can the authors discuss the generalizability of this study? What % of participants came from different suburban areas other than Polana Caniço? This is a relevant issue in this study as if I understood the majority o participants are from Polana Caniço area. R: We thank you for the comments. Data will be used in another manuscript (incidence paper).”

R: Our sincere apologies for not fully addressing the generalizability comment in the previous revision. We have now addressed the comment as suggested by including a paragraph on the generalizability of the study under a new ‘Limitations’ section of the manuscript. 

3. The third issue is acknowledging the self-presentation bias that another reviewer (#3) had raised but was not addressed by the authors. I also state that these type of studies are highly influenced by social desirability bias—this need to be considered in the limitations and actions to minimise it stated.

R: We appreciate your comment, and our apologies for the oversight. The issue pertaining to self-presentation bias as well as social desirability bias have been addressed under the new ‘Limitations’ section of the manuscript.

---

## [Editor Report · Decision Letter 2]

4 Nov 2021

Young at risk-people in Maputo City, Mozambique, present a high willingness to participate in HIV trials: results from an HIV vaccine preparedness cohort study

PONE-D-20-31594R2

Dear Dr. Capitine,

We are pleased to inform you that your manuscript has been judged potentially scientifically suitable for publication and will be formally accepted for publication once it meets all outstanding technical requirements. Please kindly note two final proposed comments from the Academic Editor.

Kind regards,

Karine Dubé, DrPH

Academic Editor

PLOS ONE

Additional Editor Comments:

We strongly encourage the authors to cite previous similar research in Mozambique ()https://pubmed.ncbi.nlm.nih.gov/25555105/). The authors should adhere to the NIAID Language Guide (https://www.hptn.org/resources/HIVLanguageGuide). The expression 'HIV-infected' and 'HIV-uninfected' are now considered stigmatizing. We encourage person-first language as much as possible (e.g., people with HIV, people without HIV).

Reviewers' comments:

N/A

---

## [Editor Report · Acceptance letter]

19 Nov 2021

PONE-D-20-31594R2 

Young at risk-people in Maputo City, Mozambique, present a high willingness to participate in HIV trials: results from an HIV vaccine preparedness cohort study 

Dear Dr. Capitine:

I'm pleased to inform you that your manuscript has been deemed suitable for publication in PLOS ONE. Congratulations! Your manuscript is now with our production department. 

Kind regards, 

on behalf of

Dr. Karine Dubé 

Academic Editor

PLOS ONE